# From Neural Networks to Logical Theories: The Correspondence between Fibring Modal Logics and Fibring Neural Networks

**Ouns El Harzli**[1,2,3*]   **Bernardo Cuenca Grau**[1]   **Artur D'Avila Garcez**[4]

**Ian Horrocks**[1]   **Tarek R. Besold**[2]

[1]Department of Computer Science, University of Oxford, UK
[2]Sony AI Barcelona, Spain
[3]Sony AI Tokyo, Japan
[4]City St George's, University of London, UK

## Abstract

Fibring of modal logics is a well-established formalism for combining countable families of modal logics into a single fibred language with common semantics, characterized by fibred models. Inspired by this formalism, fibring of neural networks was introduced as a neurosymbolic framework for combining learning and reasoning in neural networks. Fibring of neural networks uses the (pre-)activations of a trained network to evaluate a fibring function computing the weights of another network whose outputs are injected back into the original network. However, the exact correspondence between fibring of neural networks and fibring of modal logics was never formally established. In this paper, we close this gap by formalizing the idea of fibred models *compatible* with fibred neural networks. Using this correspondence, we then derive non-uniform logical expressiveness results for Graph Neural Networks (GNNs), Graph Attention Networks (GATs) and Transformer encoders. Longer-term, the goal of this paper is to open the way for the use of fibring as a formalism for interpreting the logical theories learnt by neural networks with the tools of computational logic.

## 1 Introduction

The advent of large language models has created unprecedented interest in the task of reasoning in neural networks. Logical reasoning is arguably the best perspective to study and develop this capability in neural networks. Logic offers precise definitions, validity conditions and a formalism that is amenable to verification. As a result, there has been a surge of interest in the field of neurosymbolic AI that studies the integration of neural networks with logical reasoning (Besold et al., 2022; Lamb et al., 2020; d'Avila Garcez & Lamb, 2023).

Fibring of neural networks (Garcez & Gabbay, 2004) is a theoretical concept from the neurosymbolic AI literature introduced as a way of combining neural network architectures for learning and reasoning. The idea of fibring is to make the parameters of a network a function of another network, with the resulting output injected back into the original network when computing the output of the combined (fibred) neural network. This theoretical framework was initially inspired by the concept of fibring logics (Gabbay, 1999), in particular combining modal logics (Chellas, 1980), which play an important role in systems verification; temporal logic, a special case of modal logic, is extensively used in formal verification (Vardi & Wolper, 1986). Fibred neural networks were shown in (Garcez & Gabbay, 2004) to be strictly more expressive than the usual composition of neural networks. Fibred networks intended to offer a framework for the study of the combination of learning and reasoning in neural

---

*Correspondence to: `ouns.el.harzli@cs.ox.ac.uk`

networks, whereby one network's learning influences another network's inference. In a fibred modal logic language, the Kripke model[1] and the possible world in which a formula is evaluated are a function of a possible world in another Kripke model. However, the precise correspondence between fibring of neural networks and fibring of modal logics was not formally established until now.

An important research area in neurosymbolic AI aims to establish correspondences between logic and modern neural architectures, motivated by the prospect of rendering the latter more reliable, interpretable or verifiable. In particular, Graph Neural Networks (GNNs) and Transformers have become essential components in contemporary machine learning, each tackling specific but sometimes intersecting challenges across a wide range of applications. GNNs excel at processing structured graph data, having found extensive use in fields such as social network analysis, drug discovery, and knowledge graphs (Salamat et al., 2021; Xiong et al., 2021; Ye et al., 2022). Transformers have revolutionized natural language processing by modeling intricate contextual relationships in sequential data, providing unprecedented capabilities for tasks such as language understanding and generation (Vaswani et al., 2017; Devlin et al., 2019; Brown et al., 2020; OpenAI, 2023; Bronstein et al., 2021). The capabilities of these architectures have sparked a significant research effort to analyze rigorously their *logical expressiveness*, i.e. the classes of formal languages that they can compute, and their formal verification. An interesting connection exists between GNNs, and Transformer encoders (Bronstein et al., 2021): Transformer encoders can be viewed as GNNs applied to complete graphs with added positional encoding and an attention mechanism. This connection, however, has not been exploited in the literature on logical expressivity and formal verification. This gap leaves potential for unifying logical characterizations of both architectures.

**Logical expressiveness of modern architectures.** Existing logical expressiveness results for GNNs and Transformers can be categorized into: *discriminative power* (e.g. is there a logical classifier that can distinguish the same pairs of nodes as a GNN?); *uniform expressiveness* (e.g. is there a logical formula whose truth values coincide with the output of a GNN given *any* input graph and node?); and *non-uniform expressiveness* (in which the logical formulas can depend on the input). In this categorization, uniform expressiveness stand out as the most powerful kind, providing complete logical characterization for neural architectures (i.e. providing a single logical formula for each instance (choice of weights) of a neural architecture independently of the input). Initial results established that the discriminative power of standard GNNs is upper-bounded by the Weisfeiler-Leman (WL) test (Barceló et al., 2020). Subsequently, Grohe (2023) established non-uniform expressiveness results for GNNs in terms of Boolean circuits and descriptive complexity. More recently, uniform expressiveness results of broad classes of GNNs have been derived (Nunn et al., 2024; Benedikt et al., 2024) using known logics called Presburger logics, which involve counting modalities and linear inequalities. Using existing results about the satisfiability of such logics, results about the computational complexity of formal verification problems for neural networks follow as corollaries (e.g. answering the question: given an output and a GNN, is there an input to the GNN that yields that output?). In particular, Benedikt et al. (2024) provides a taxonomy for the decidability of these verification problems depending on the types of aggregations and activation functions being used. Cuenca Grau et al. (2025) established uniform expressiveness results for bounded GNNs in terms of fragments of first-order logic, showcasing how restrictive yet practical assumptions on the class of GNNs may simplify significantly their logical characterization. Transformer encoders have also been studied through the lens of circuit complexity theory. Unique Hard Attention Transformers (UHATs) have been mapped to fragments of the complexity class of $AC^0$ languages with extensions based on a restricted form of first-order logic (Hao et al., 2022). Average Hard Attention Transformers (AHATs) have been shown capable of capturing more complex languages, including those outside $AC^0$, and more recently a uniform lower bound has been established w.r.t. linear temporal logic using a language called LTL (C, +), which also involves counting modalities (Barcelo et al., 2024). While expressiveness results for GNNs and Transformers share certain similarities (e.g. counting modalities), there is still no unified theory that accounts for both architectures.

**Contributions.** In this paper, we propose fibring of modal logics as a new formalism to study the logical expressiveness of neural architectures, including GNNs and Transformers. We start by redefining fibring neural networks (Garcez & Gabbay, 2004) to make its use easier in the context

---

[1]The word *model* here refers to logic structures with assignments of truth-values *true* or *false*, differently from the use of the word in neural networks.

of the above literature. We then formally establish an exact correspondence between fibring neural networks and fibring modal logics. That is, (i) we define a fibred language based on a fibring architecture; (ii) we introduce the notion of *compatible* fibred models (with fibred neural networks and their inputs) on which to interpret the formulas; (iii) we prove that our proposed fibred logic is a valid fibred logic by demonstrating that the class of compatible fibred models is non-empty and closed under Kripke-model isomorphism; and (iv) we construct the formulas from our fibred logic whose truth values coincide with the outputs of fibred neural networks. Subsequently, we prove that fibred neural networks can be used to non-uniformly describe large classes of GNNs, Graph Attention Networks (GATs) and Transformer encoder architectures. It follows that the corresponding fibred languages provide non-uniform expressiveness results for these network architectures. Our hope is that fibring can become a unifying formalism for the study of GNNs and Transformers in neurosymbolic AI.

As future directions of research, we speculate that fibring as a formalism might hold the key to deriving and unifying uniform expressiveness results for GNNs and Transformer encoders, and we present our main arguments for this possible unification in Section 6. We argue that the proposed fibring framework provides a new intuitive way of thinking about GNNs and Transformers as successive combinations of underlying modal logics, and we discuss how this new perspective might be useful for the comparative evaluations, interpretability and verification of modern neural networks.

The remainder of the paper is organized as follows. Section 2 introduces the notation used throughout the paper. Section 3 provides the new definition of fibring. Section 4 proves the correspondence between fibring networks and modal logics. Section 5 applies this result to derive expressiveness results for GNNs, GATs and Transformer Encoders. Section 6 concludes the paper and discusses directions for future work.

## 2 Preliminaries

We work with vectors and matrices of rational numbers. The $i$th entry of a vector $\mathbf{v}$ is written $v_i$, and the entry in row $i$, column $j$ of a matrix $\mathbf{A}$ is written $A_{i,j}$. If two matrices have the same number of columns, we can place them side by side to form a larger matrix, called their concatenation.

A *neural architecture* $\mathbb{A}$ is specified by the number of its layers $L$, the dimension of each layer $d_\ell$, and an activation map $\sigma^\ell : \mathbb{Q}^{d_\ell} \mapsto \mathbb{Q}^{d_\ell}$ for each hidden layer. With $L$ layers, $d_0$ and $d_L$ are referred to as the input and output dimensions, respectively, and for each hidden layer $\ell$, the activation map $\sigma^\ell$ is computable in polynomial time. We may also talk about a portion of the architecture ranging from layer $p$ to layer $q$, and call this the *sub-architecture* $\mathbb{A}^{p:q}$. If $q = L$, we write $\mathbb{A}^{p\uparrow}$ for simplicity.

A *network instance* of an architecture assigns weights and biases, known as the set of parameters, to each layer. The weights of layer $\ell$ form a matrix $\mathbf{W}^\ell \in \mathbb{Q}^{d_\ell \times d_{\ell-1}}$, and the biases form a vector $\mathbf{b}^\ell \in \mathbb{Q}^{d_\ell}$. To apply a network instance $\mathcal{N}$ to an input vector $x$, we compute the weighted sum of the input weighted by the parameters followed by the application of the activation function in the usual way proceeding from the input layer to the output. The final output of the network is a vector of the output's dimension. Formally, the application of $\mathcal{N}$ to $\mathbf{x} \in \mathbb{Q}^{d_0}$ generates a sequence $\mathbf{h}^1, \ldots, \mathbf{h}^L$ of vectors defined as $\mathbf{h}^\ell = \mathbf{W}^\ell \cdot \mathbf{x}^{\ell-1} + \mathbf{b}^\ell$, where $\mathbf{x}^0 = \mathbf{x}$, $\mathbf{x}^\ell = \sigma^\ell(\mathbf{h}^\ell)$. The result $\mathcal{N}(\mathbf{x})$ of applying $\mathcal{N}$ to $\mathbf{x}$ is the vector $\mathbf{h}^L$. We also talk about *sub-networks*. The sub-network $\mathcal{N}^{p:q}$ consists of the layers from $p$ through $q$ and parameters. If $q = L$, we write $\mathcal{N}^{p\uparrow}$.

## 3 Fibring Neural Networks

In this section, we introduce a definition of fibred neural networks which generalizes the original definition of Garcez & Gabbay (2004) to any number and possible combinations of neural networks.

A *fibring architecture* is a directed tree $\mathcal{F}$ whose nodes $v$ are labeled with neural architectures $\mathbb{A}_v$. Each edge $(v, v')$ is labeled with: (i) a layer number $\ell$ in the parent architecture; and (ii) a set of positions $S$ in that layer denoting the fibred neurons. We impose the additional requirement that for any two edges $(v, w)$ and $(v, u)$ sharing the parent node and labeled with the same layer number, the corresponding sets of positions must be disjoint. Finally, we denote by $\mathscr{F}$ the class of fibring architectures which verify the property that: (i) the architecture at the root has two linear layers with

input dimension $n$ and output dimension 1, and (ii) every edge leaving the root node is labeled with the same layer number $\ell$.

A *fibred network* $\widetilde{\mathcal{N}}$ is a tuple $\langle \mathcal{N}, \mathcal{F}, \tilde{f} \rangle$ where $\mathcal{N}$ is a network instance, $\mathcal{F}$ is a fibring architecture, and $\tilde{f}$ is a finite collection of neural *fibring functions* matching $\mathcal{F}$, i.e. one function $\tilde{f}_{(v,v')}$ for each edge $(v, v')$ in $\mathcal{F}$, which specifies how to build an instance of the child architecture and what input to give it; specifically, for an edge $(v, v')$ labeled with layer number $\ell$, the fibring function $\tilde{f}_{(v,v')}$ maps vectors of dimension $\le d_\ell$ to a network instance of architecture $\mathbb{A}_{v'}$ and a valid input vector for that architecture.

To apply a fibred neural network $\widetilde{\mathcal{N}}$ to an input $\mathbf{x}$, we start at the root. Whenever we reach a layer that has edges leading to children, we pause, call the corresponding fibring function, and pass part of the current vector into the child network. The child network produces a new vector, which is spliced back into the parent's computation at the specified positions. The overall computation proceeds recursively until the output layer of the root is reached.

Formally, the computation is defined inductively. If $\mathcal{F}$ has only the root node $u$, then $\widetilde{\mathcal{N}}(\mathbf{x}) = \mathcal{N}(\mathbf{x})$. Otherwise, let $u_1, \dots, u_k$ be the children of $u$, with edges $(u, u_i)$ labeled $(\ell_i, S_i)$, let $\mathcal{F}_i$ be the subtree rooted at $u_i$, and let $\tilde{f}_i$ be the restriction of $\tilde{f}$ to edges in $\mathcal{F}_i$. Assume the children are ordered so that $\ell_1 \le \ell_2 \le \dots \le \ell_k$. For each stage $1 \le i \le k$, define a tuple $(\mathbf{x}_i, \mathcal{N}_i, \mathbf{y}_i, \mathbf{h}_i)$ as follows:

$$\mathbf{x}_i = \begin{cases} \mathcal{N}^{1:\ell_1}(\mathbf{x}), & i = 1, \\ \mathcal{N}^{\ell_{i-1}:\ell_i}(\mathbf{h}_{i-1}), & i > 1, \end{cases}$$

$$(\mathcal{N}_i, \mathbf{y}_i) = \tilde{f}_{(u,u_i)}(\mathbf{x}_i),$$

$\mathbf{h}_i = \mathbf{x}_i$ with entries in $S_i$ replaced by $\langle \mathcal{N}_i, \mathcal{F}_i, \tilde{f}_i \rangle(\mathbf{y}_i)$, the application of the fibred network to $\mathbf{y}_i$.

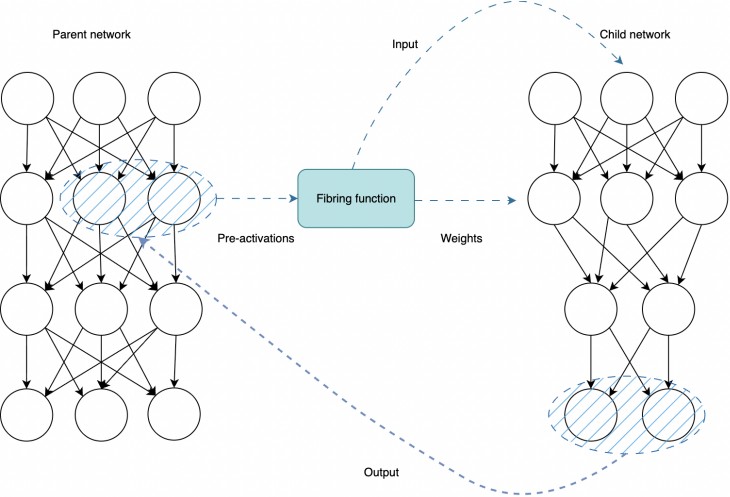

Figure 1: Illustration of fibring between two architectures. The pre-activations at a given layer within the parent network are fed into a fibring function to produce the input and the weights of the child network, the output of the child network is then reinjected at the same pre-activations.

The final output is:

$$\widetilde{\mathcal{N}}(\mathbf{x}) = \mathcal{N}^{\ell_k \uparrow}(\mathbf{h}_k).$$

If the output of the root network is a scalar, we can interpret the fibred network as a classifier in the usual way: on input $\mathbf{x}$, it outputs *true* if the final value is strictly greater than 0, and *false* otherwise.

## 4 EXACT CORRESPONDENCE BETWEEN FIBRING LOGICS AND FIBRING NETWORKS

In this section, we introduce a fibred modal language for combining different modal systems and show it captures exactly the behavior of fibred neural networks.

**Modal logics and fibring (Gabbay, 1999):** Fix a finite set of propositions $PROP = \{p_1, \ldots, p_n\}$ and a countable collection of modal operators $\Box_i$ (one for each index $i$). For each $i$, logic $\mathcal{L}_i$ has formulas built from propositions, $\top$ (true), Boolean connectives, and the modal operators $\Box_i$. The semantics is the usual Kripke semantics: each $\mathcal{L}_i$ uses a class of Kripke models, and $\Box_i\varphi$ holds at a world $w$ in a model when $\varphi$ holds at all accessible worlds $w_i$ from $w$ according to a pre-defined *accessibility relation* $R$ such that $R(w_i, w)$ holds [2].

To *fibre* these logics, we allow all the modal operators $\Box_i$ in one combined language. That is, formulas are built from the grammar:

$$\varphi ::= p \mid \top \mid \varphi_1 \wedge \varphi_2 \mid \neg\varphi \mid \Box_i\varphi \tag{1}$$

where $p \in PROP$ and $i$ ranges over the fibred modal logics $\mathcal{L}_i$.

A *fibred model* chooses, for each $i$, one Kripke model for $\mathcal{L}_i$, and also provides a way of *jumping* from worlds of other models into the model for $i$. Intuitively, evaluating $\Box_j\varphi$ at a world in model $i$ either uses the native accessibility for the model $i$ if $j = i$, or first jumps into model $j$ and then evaluates there.[3] Formally, a fibred model $\mathcal{M}$ consists of one Kripke model $\mathbf{m}_i$ for each logic $\mathcal{L}_i$, together with a family of logical *fibring functions* $f_i$. Each $f_i$ maps each world in another model $\mathbf{m}_j$ ($j \neq i$) to a world in $\mathbf{m}_i$, while $f_i$ tells us how to jump from another component into the $i$-th one.

Satisfaction in a fibred model $\mathcal{M}$ is defined as follows. If $w$ is a world in $\mathbf{m}_i$, then:

- $\mathcal{M}, w \models p$ iff $w$ is in the valuation of $p$ in $\mathbf{m}_i$.
- $\mathcal{M}, w \models \top$ always.
- $\mathcal{M}, w \models \varphi_1 \wedge \varphi_2$ iff $\mathcal{M}, w \models \varphi_1$ and $\mathcal{M}, w \models \varphi_2$.
- $\mathcal{M}, w \models \neg\varphi$ iff not $\mathcal{M}, w \models \varphi$.
- $\mathcal{M}, w \models \Box_j\varphi$ iff *(i)* $j = i$ and for all $w'$ with $(w, w')$ in the accessibility relation of $\mathbf{m}_i$, we have $\mathcal{M}, w' \models \varphi$; or *(ii)* $j \neq i$ and $\mathcal{M}, f_j(w) \models \Box_j\varphi$.

**A fibred logic for a fibring architecture:** Let $\mathcal{F}$ be a fibring architecture. For each node $v$ in $\mathcal{F}$ and each layer number $\ell$ of $\mathbb{A}_v$, associate a distinct modal operator $\Box_{v,\ell}$, interpreted over the class of all finite Kripke models; let $\mathcal{L}$ be the resulting fibred logic. Based on $\mathcal{L}$ we will define another fibred logic $\mathcal{L}_\mathcal{F}$ using the notion of *compatible* models, defined next.

**Definition 4.1.** Let $\mathbf{m}$ be a Kripke model and $\mathcal{N}$ a network instance. A map $\pi$ that assigns to each world of $\mathbf{m}$ an input vector for $\mathcal{N}$ is $(\mathbf{m}, \mathcal{N})$–*admissible* if it is injective and, for all worlds $w, w'$,

$$w \text{ is related to } w' \text{ in } \mathbf{m} \iff \mathcal{N}(\pi(w)) = \mathcal{N}(\pi(w')).$$

**Definition 4.2.** Let $\mathcal{M}$ be a fibred model of $\mathcal{L}$ and denote with $\mathbf{m}_{v,\ell}$ the component Kripke model in $\mathcal{M}$ for node $v$ in $\mathcal{F}$ and layer number $\ell$ in $\mathbb{A}_v$. Let $\widetilde{\mathcal{N}} = \langle \mathcal{N}, \mathcal{F}, \tilde{f} \rangle$, let $\mathbf{x} \in \{0, 1\}^n$, and let $u$ be the root of $\mathcal{F}$. We say that $\mathcal{M}$ is *compatible* with $(\widetilde{\mathcal{N}}, \mathbf{x})$ if we can assign:

- to each node $v$ in $\mathcal{F}$, a network instance $\mathcal{N}_v$ of $\mathbb{A}_v$, and

- to each pair $(v, \ell)$ of a node in $\mathcal{F}$ and a layer $\ell$ in $\mathbb{A}_v$, a bijection $\pi_{v,\ell}$ from a finite set of Kripke worlds to a finite set of vectors,

such that the following compatibility conditions hold:

---

[2]In a Kripke model, the accessibility relation can be any relation on the set of worlds, i.e. fully determined by a set of pairs of worlds.

[3]We assume that the classes of models are disjoint so that each world's *home* model is unambiguous and we also assume that all component Kripke models are finite.

(C0) $\mathcal{N}_u = \mathcal{N}$ and $\pi_{u,1}$ maps each world $w$ in $\mathbf{m}_{u,1}$ to the vector in $\{0,1\}^n$ whose $i$-th bit is 1 exactly when proposition $p_i$ is true at $w$.

(C1) $\pi_{v,\ell}$ is $(\mathbf{m}_{v,\ell}, \mathcal{N}_v^{\ell\uparrow})$-admissible.

(C2) Assume $v$ has children $v_1, \ldots, v_k$, ordered by the layer numbers $\ell_1, ..., \ell_k$ labeling edges $(v, v_i)$, and let $w$ be the world reached via the composition of logical fibring functions from $(\pi_{u,1})^{-1}(\mathbf{x})$ to Kripke model $\mathbf{m}_{v,1}$. Running $\langle \mathcal{N}_v, \mathcal{F}_v, \tilde{f}_v \rangle$ on $\pi_{v,1}(w)$ produces tuples $(\mathbf{x}_i, \mathcal{N}_i, \mathbf{y}_i, \mathbf{h}_i)$ for $1 \leq i \leq k$. Then:

   1. $\mathcal{N}_{v_i} = \mathcal{N}_i$ and $\mathbf{y}_i = \pi_{v_i,1}(f_{v_i,1}(w))$.
   2. $\mathbf{h}_i = \pi_{v,\ell_i}(f_{v,\ell_i}(w))$.
   3. The set of propositions that are true at world $\pi_{v,\ell_k}^{-1}(\mathbf{h}_k)$ in model $\mathbf{m}_{v,\ell_k}$ agrees with the set of propositions that are true at $\pi_{v_i,1}^{-1}(\mathbf{y}_i)$ in model $\mathbf{m}_{v_i,1}$ for $1 \leq i \leq k$.

For a fixed pair $(v, \ell)$ of a node $v$ in $\mathcal{F}$ and a layer number $\ell$ in $\mathbb{A}_v$, let:

$$\mathsf{Comp}_{\mathcal{F}}(v, \ell) := \left\{ \mathbf{m} \;\middle|\; \begin{array}{l} \exists \widetilde{\mathcal{N}}, \mathbf{x}, \mathcal{M} \text{ a fibred model of } \mathcal{L} \text{ compatible with } (\widetilde{\mathcal{N}}, \mathbf{x}) \\ \text{such that the } (v, \ell)\text{-component of } \mathcal{M} \text{ is } \mathbf{m} \end{array} \right\}.$$

In words, $\mathsf{Comp}_{\mathcal{F}}(v, \ell)$ is the *projection* of all compatible fibred models onto their $(v, \ell)$ component.

**Proposition 4.3.** *The class $\mathsf{Comp}_{\mathcal{F}}(v, \ell)$ is non-empty and closed under Kripke-model isomorphism.*

*Proof. Non-empty.* The class $\mathsf{Comp}_{\mathcal{F}}(v, \ell)$ contains at least the Kripke models of fibred models obtained by explicitly running the computation of fibred networks $\tilde{\mathcal{N}}$ on vectors $\mathbf{x}$.

Specifically, for each $\tilde{\mathcal{N}} = \langle \mathcal{N}, \mathcal{F}, \tilde{f} \rangle$ and $\mathbf{x}$, we can construct a compatible fibred model as follows.

We start from the root model: the domain of the root model contains a distinct element $w_{\mathbf{z}}$ for each vector $\mathbf{z} \in \{0,1\}^n$, two worlds $w_{\mathbf{z}}, w_{\mathbf{z}'}$ are related iff $\mathcal{N}(\mathbf{z}) = \mathcal{N}(\mathbf{z}')$, and the valuation is given by: $p_j$ holds at world $w_{\mathbf{z}}$ in the root model iff $z_j = 1$.

We run the fibred neural network $\tilde{\mathcal{N}}$ on all vectors $\mathbf{z} \in \{0,1\}^n$ (recursively generating sequences of tuples $(\mathbf{x}_i, \mathcal{N}_i, \mathbf{y}_i, \mathbf{h}_i)$) and we collect at each $(v, \ell)$ all vectors attained by the computations, that is, for each $(v, \ell_i)$ we collect vectors $\mathbf{h}_i$ and for each $(v_i, 1)$, we collect vectors $\mathbf{y}_i$. Furthermore, we run the fibred neural network $\tilde{\mathcal{N}}$ on the specific vector $\mathbf{x}$ and we collect at each $v_i$, the network instance $\mathcal{N}_i$ attained by the computation.

We then define Kripke models as follows: the domain of $\mathbf{m}_{v,\ell}$ contains a distinct element $w_{\mathbf{v}}$ for each vector $\mathbf{v}$ collected at $(v, \ell)$, two worlds $w_{\mathbf{v}}, w_{\mathbf{v}'}$ are related iff $\mathcal{N}_v^{\ell\uparrow}(\mathbf{v}) = \mathcal{N}_v^{\ell\uparrow}(\mathbf{v}')$ where $\mathcal{N}_v$ is the network instance collected at $v$.

The valuations are defined recursively starting from the root model: a proposition $p$ holds at world $w_{\mathbf{h}}$ in model $\mathbf{m}_{v,\ell_i}$ iff there is a world $w_{\mathbf{v}}$ in $\mathbf{m}_{v,1}$ such that $\mathbf{h} = \mathbf{h}_i$ is attained by the computation of $\langle \mathcal{N}_v, \mathcal{F}_v, \tilde{f}_v \rangle(\mathbf{v})$ and $p$ holds at $w_{\mathbf{v}}$ in model $\mathbf{m}_{v,1}$; and a proposition $p$ holds at world $w_{\mathbf{y}}$ in model $\mathbf{m}_{v_i,1}$ iff there is a world $w_{\mathbf{v}}$ in $\mathbf{m}_{v,1}$ such that $\mathbf{y} = \mathbf{y}_i$ is attained by the computation of $\langle \mathcal{N}_v, \mathcal{F}_v, \tilde{f}_v \rangle(\mathbf{v})$ and $p$ holds at $w_{\mathbf{h}_k}$ in model $\mathbf{m}_{v,\ell_k}$.

Finally, we require that the (logical) fibring function from $\mathbf{m}_{v,1}$ to $\mathbf{m}_{v,\ell_i}$ verifies for each world $w_{\mathbf{v}}$ in $\mathbf{m}_{v,1}$, $f_{v,\ell_i}(w_{\mathbf{v}}) = w_{\mathbf{h}_i}$ where $\mathbf{h}_i$ is the vector attained by the computation of $\langle \mathcal{N}_v, \mathcal{F}_v, \tilde{f}_v \rangle(\mathbf{v})$; and the (logical) fibring function from $\mathbf{m}_{v,1}$ to $\mathbf{m}_{v_i,1}$ verifies, for each world $w_{\mathbf{v}}$ in $\mathbf{m}_{v,1}$, $f_{v_i,1}(w_{\mathbf{v}}) = w_{\mathbf{y}_i}$ where $\mathbf{y}_i$ is the vector attained by the computation of $\langle \mathcal{N}_v, \mathcal{F}_v, \tilde{f}_v \rangle(\mathbf{v})$.

The above proof construction covers all compatibility conditions and ensures that the resulting fibred model is compatible with $(\widetilde{\mathcal{N}}, \mathbf{x})$ by taking the bijections $\pi_{v,\ell} : w_{\mathbf{v}} \mapsto \mathbf{v}$.

*Closed under Kripke-model isomorphism.* Let $\mathbf{m} \in \mathsf{Comp}_{\mathcal{F}}(v, \ell)$ come from some compatible $\mathcal{M}$, and let $\pi : \mathbf{m} \cong \mathbf{m}'$ be an isomorphism. Form a new fibred model $\mathcal{M}'$ by replacing the $(v, \ell)$-component of $\mathcal{M}$ with $\mathbf{m}'$ and replace the bijection $\pi_{v,\ell}$ by $\pi_{v,\ell} \circ \pi^{-1}$. This preserves (C0)–(C2): edges, valuations, and fibring jumps are transported through $\pi$ without changing any observable behavior. Hence $\mathcal{M}'$ is still compatible with the same $(\widetilde{\mathcal{N}}, \mathbf{x})$, so $\mathbf{m}' \in \mathsf{Comp}_{\mathcal{F}}(v, \ell)$. $\square$

This shows that the subclass of compatible models is a valid fibred logic, since the projections of compatible models represent a non-empty class of Kripke models closed under isomorphism (hence a reasonable class of Kripke models). We refer to this fibred logic as $\mathcal{L}_{\mathcal{F}}$.

**Equivalence between fibring neural networks and a fragment of fibred logic:**

**Definition 4.4.** Let $\mathcal{F}$ be a fibring architecture and let $\varphi$ be a propositional formula. The formula $\psi(\varphi, \mathcal{F}) \in \mathcal{L}_{\mathcal{F}}$ is recursively defined as follows:

- If $\mathcal{F}$ has only a single node, then $\psi(\varphi, \mathcal{F}) = \varphi$.

- Otherwise, let $v_1, \ldots, v_k$ be the children of the root, with corresponding subtrees $\mathcal{F}_1, \ldots, \mathcal{F}_k$. For each $i$, let $l_i$ be the largest layer label on an edge leaving $v_i$ (or $l_i = 1$ if none). Then set

$$\psi(\varphi, \mathcal{F}) = \Box_{v_1, l_1} \psi(\varphi, \mathcal{F}_1) \wedge \cdots \wedge \Box_{v_k, l_k} \psi(\varphi, \mathcal{F}_k).$$

**Theorem 4.5.** *Let $\mathcal{F} \in \mathscr{F}$ be a fibring architecture rooted at $u$. For every network instance $\mathcal{N}$ of the root architecture $\mathbb{A}_u$, there is a propositional formula $\varphi$ such that, for every input $\mathbf{x} \in \{0,1\}^n$, for every collection of fibring functions $\tilde{f}$ matching $\mathcal{F}$, and every fibred model $\mathcal{M}$ compatible with $\langle \mathcal{N}, \mathcal{F}, \tilde{f} \rangle$ and $\mathbf{x}$, we have that the following holds:*

$$\mathcal{M}, (\pi_{u,1})^{-1}(\mathbf{x}) \models \psi(\varphi, \mathcal{F}) \quad \textit{iff} \quad \langle \mathcal{N}, \mathcal{F}, \tilde{f} \rangle \textit{ classifies } \mathbf{x} \textit{ as True.}$$

*Proof idea.* In a compatible fibred model, the accessibility relations between worlds mirror the behavior of network instances on their inputs (condition C1), and the fibring functions on worlds mirror the way a parent network delegates part of its computation to children (condition C2). Thus evaluating the fibred network on $\mathbf{x}$ corresponds exactly to checking the truth value of the formula $\psi(\varphi, \mathcal{F})$ at the matching world, since both follow the same recursive structure given in Definition 4.4.

*Proof.* Fix a network instance $\mathcal{N}$ of the root architecture. Define the *characteristic formula* of $\mathcal{N}$ by

$$\varphi := \bigvee_{\mathbf{h} \in \{0,1\}^n : \mathcal{N}(\mathbf{h}) > 0} \left( \bigwedge_{h_k = 1} p_k \wedge \bigwedge_{h_k = 0} \neg p_k \right).$$

Fix $\mathcal{F} \in \mathscr{F}$, $\mathbf{x} \in \{0,1\}^n$ and $\mathcal{M}$ compatible with $\langle \mathcal{N}, \mathcal{F} \rangle$ and $\mathbf{x}$. We prove by induction on the depth of $\mathcal{F}$ that, for the world $w = (\pi_{u,1})^{-1}(\mathbf{x})$ at the root: $\mathcal{M}, w \models \psi(\varphi, \mathcal{F})$ iff $\langle \mathcal{N}, \mathcal{F}, \tilde{f} \rangle$ classifies $\mathbf{x}$ as true.

*Base case (leaf).* If $\mathcal{F}$ has only the root then $\psi(\varphi, \mathcal{F}) = \varphi$ and by (C0), at the root model $\mathbf{m}_{u,1}$, $\varphi$ satisfies: $\mathbf{m}_{u,1}, \pi_{u,1}{}^{-1}(\mathbf{h}) \models \varphi$ iff $\mathcal{N}(\mathbf{h}) > 0, \mathbf{h} \in \{0,1\}^n$.

*Induction step.* Suppose the root has children $u_1, \ldots, u_k$ with labels $(\ell_i, S_i)$, subtrees $\mathcal{F}_i$ and $\tilde{f}_i$ restrictions of $f$ to $\mathcal{F}_i$. Run the root until layer $\ell_1$, obtain $\mathbf{x}_1$; fibre to $u_1$ by $(\mathcal{N}_1, \mathbf{y}_1) = \tilde{f}_{(u, u_1)}(\mathbf{x}_1)$; run the child fibred network $\langle \mathcal{N}_1, \mathcal{F}_1, \tilde{f}_1 \rangle$ on $\mathbf{y}_1$; splice its output back to get $\mathbf{h}_1$; and continue similarly for $i = 2, \ldots, k$. By (C1)–(C2) and the semantics of $\Box_{v, \ell}$, evaluating $\psi(\varphi, \mathcal{F}) = \bigwedge_i \Box_{v_i, \ell_i} \psi(\varphi, \mathcal{F}_i)$ at $w$ amounts to evaluating each $\psi(\varphi, \mathcal{F}_i)$ at the world reached by the corresponding fibring jump. By the induction hypothesis, each of those subformulas is true exactly when the corresponding child computation returns the vector that is spliced into the parent. Hence $\mathcal{M}, w \models \psi(\varphi, \mathcal{F})$ iff the overall fibred computation yields a final root output $> 0$, i.e., iff $\langle \mathcal{N}, \mathcal{F}, \tilde{f} \rangle$ classifies $\mathbf{x}$ as true. $\square$

## 5 APPLICATION TO NON-UNIFORM EXPRESSIVENESS OF GNNS, GATS AND TRANSFORMER ENCODERS

**Graph Neural Networks.** A Graph Neural Network (GNN) for $\mathbb{A}$ is a tuple $\mathcal{G} = \langle \{\mathbf{A}^\ell\}_{\ell=1}^L, \{\mathbf{B}^\ell\}_{\ell=1}^L, \{\mathbf{b}^\ell\}_{\ell=1}^L \rangle$. For each layer $\ell$, matrices $\mathbf{A}^\ell \in \mathbb{Q}^{d_\ell \times d_{\ell-1}}$ and $\mathbf{B}^\ell \in \mathbb{Q}^{d_\ell \times d_{\ell-1}}$ are weight matrices and vector $\mathbf{b}^\ell \in \mathbb{Q}^{d_\ell}$ is a bias vector. A Graph Attention Network (GAT) $\mathcal{G}'$ for $\mathbb{A}$ contains all the elements of a GNN plus an additional collection $\{\mathbf{a}^\ell\}_{\ell=1}^L$ of attention vectors, where $\mathbf{a}^\ell \in \mathbb{Q}^{2 \cdot d_\ell}$. Both GNNs and GATs apply to undirected graphs $G = (V, E)$ where a node vector $\mathbf{x}_u \in \{0,1\}^{d_0}$ is associated to each node $u \in V$ and where the neighborhood of node $u$ is defined

as $\mathrm{N}(u) = \{v \in V : \{u, v\} \in E\}$. Their application to graph $G$ with node vectors $\boldsymbol{x} := (\mathbf{x}_u)_{u \in V}$ yields a sequence $(\mathbf{x}_u^1)_{u \in V}, \ldots, (\mathbf{x}_u^L)_{u \in V}$ of node vectors and the final result is a graph with the same nodes and edges as $G$ but with updated node vectors $\{\mathbf{x}_u^L\}_{u \in V}$. When there is no ambiguity, we will refer to $\mathbf{x}_u^L$ as $\mathcal{G}(G, \boldsymbol{x}, u)$, the result at node $u$ of applying GNN (or GAT) $\mathcal{G}$ to graph $G$ with node features $\boldsymbol{x}$.

In the application of GNN $\mathcal{G}$ to $G, \boldsymbol{x}$ is defined, for each $u \in V$, as $\mathbf{x}_u^\ell = \sigma^\ell(\mathbf{h}_u^\ell)$, where $\mathbf{x}_u^0 = \mathbf{x}_u$ and $\mathbf{h}_u^\ell = \mathbf{B}^\ell \cdot \mathbf{x}_u^{\ell-1} + \sum_{v \in \mathrm{N}(u)} \mathbf{A}^\ell \cdot \mathbf{x}_v^{\ell-1} + \mathbf{b}^\ell$. The application of GAT $\mathcal{G}'$ to $G$ is defined analogously by replacing this expression with $\mathbf{h}_u^\ell = \alpha_{uu}^\ell \cdot \mathbf{B}^\ell \cdot \mathbf{x}_u^{\ell-1} + \sum_{v \in \mathrm{N}(u)} \alpha_{uv}^\ell \cdot \mathbf{A}^\ell \cdot \mathbf{x}_v^{\ell-1} + \mathbf{b}^\ell$, where the *attention coefficient* $\alpha_{uv}^\ell$ is the component associated to node $v$ in the following vector:

$$\mathrm{hardmax}\{\mathbf{a}^{\ell^T} \cdot (\mathbf{A}^\ell \cdot \mathbf{x}_w^{\ell-1} || \mathbf{B}^\ell \cdot \mathbf{x}_u^{\ell-1})\}_{w \in \mathrm{N}(u) \cup \{u\}}. \tag{2}$$

The hardmax function applied to a vector sets all occurrences of its largest value to the inverse of the number of its occurrences in the vector and all remaining components to $0$. The expression within the hardmax function is a vector with a component for each node $w \in \mathrm{N}(u) \cup \{u\}$ and where each component is computed as the dot product of two vectors.

**Transformer Encoders with Hard Attention.** A Transformer encoder $\mathcal{T}$ for architecture $\mathbb{A}$ is defined similarly to a GAT as $\mathcal{T} = \langle \{\mathbf{A}^\ell\}_{\ell=1}^L, \{\mathbf{B}^\ell\}_{\ell=1}^L, \{\mathbf{b}^\ell\}_{\ell=1}^L, \{\mathbf{a}^\ell\}_{\ell=1}^L \rangle$ but is applicable only to specific types of input. While GATs are applied to arbitrary graphs, Transformer encoders are restricted to complete graphs, where each node is connected to every other node (Bronstein et al., 2021). Additionally, the input graph for $\mathcal{T}$ is constructed from an ordered sequence of tokens, with each token assigned a vector representation, which is then combined with a positional encoding vector to construct a token feature. More precisely, for a complete graph with $s$ nodes associated to a token sequence $S$ of length $s$, each node is uniquely associated to a token $\xi$ at position $t \in \{0, ..., s-1\}$ in $S$ and each token is associated to a token feature $\mathbf{x}_\xi$ which is obtained as the sum $\mathbf{x}_\xi = vec(\xi) + pos(t, s)$, where $vec$ is a vector representation function mapping tokens to vectors in $\{0, 1\}^{d_0}$ and $pos$ is a positional encoding function mapping pairs of positions and sequence length to vectors in $\mathbb{Q}^{d_0}$. The application of $\mathcal{T}$ to $S$ and token features $\boldsymbol{x} := (\mathbf{x}_\xi)_{\xi \in S}$ is then obtained by applying expressions (5) and (2) to the complete graph with node features $\{\mathbf{x}_\xi\}_{\xi \in S}$. We will also note $\mathcal{T}(S, \boldsymbol{x}, \xi)$, the result at token $\xi$ of applying Transformer encoder $\mathcal{T}$ to sequence $S$ with token features $(\mathbf{x}_\xi)_{\xi \in S}$, i.e. $\mathcal{T}(S, \boldsymbol{x}, \xi)$ is the application of Transformer encoder $\mathcal{T}$ to $(S, \boldsymbol{x}, \xi)$.

**Scope of technical results.** In this paper, we restrict ourselves to GNNs and GATs with local sum aggregation, rational coefficients, and applied to Boolean vectors. Our technical results also assume that each $\sigma^\ell$ is the truncated ReLU function mapping each component $x_i$ of $\mathbf{x}$ to $\min(1, \max(0, x_i))$. We also consider GATs and Transformers with hard attention only. To avoid repetitions, the technical results in this section will be stated for GATs, but they easily extend to GNNs and Transformer encoders, by considering that a GNN is a GAT without attention, and a Transformer encoder is a GAT on complete graphs with positional encodings.

If the output dimension of the GNN or GAT (respectively, Transformer encoder) is $1$, we can interpret them as *node* (respectively, *token*) *classifiers*: on input $(G, \boldsymbol{x})$ (respectively, $(S, \boldsymbol{x})$), for each node $u$ (respectively, for each token $\xi$), it outputs *true* if $\mathbf{x}_u^L > 0$ (respectively, $\mathbf{x}_\xi^L > 0$) and *false* otherwise. By abuse of notation, we sometimes write $\mathcal{G}(G, \boldsymbol{x}, u) = $ true (or false) and the same for $\mathcal{T}(S, \boldsymbol{x}, \xi)$.

## 5.1 Fibred neural networks can non-uniformly describe GNNs, GATs and Transformer encoders

The following result establishes that fibred neural networks provide a non-uniform description of GATs, i.e. given a GAT $\mathcal{G}$, for each input $G, u, \boldsymbol{x}$ there is a different fibred neural network whose computation coincides with that of $\mathcal{G}$. Interestingly, for a given input $(G, u)$, the fibring architecture is the same, and only the fibring functions vary for different node features $\boldsymbol{x}$. This property is key to derive the (non-uniform) logical expressiveness result.

**Theorem 5.1.** *For every tuple $\tau = \langle \mathcal{G}, G, u \rangle$, where $\mathcal{G}$ is a GAT with input/output dimensions $n/1$, $G = (V, E)$ a graph, and $u \in V$, there exist a fibring architecture $\mathcal{F}^\tau$, a network instance $\mathcal{N}^\tau$ of the root architecture and a family $\mathscr{G}_\tau = \{\tilde{f}_{\boldsymbol{x}}^\tau\}_{\boldsymbol{x}}$ of collections of fibring functions matching $\mathcal{F}^\tau$, indexed*

*by possible node features $\boldsymbol{x} = (\mathbf{x}_v)_{v \in V}$, such that, for all $\boldsymbol{x}$, the fibred neural network $\langle \mathcal{N}^\tau, \mathcal{F}^\tau, \tilde{f}_{\boldsymbol{x}}^\tau \rangle$ applied to $\mathbf{x}_u$ computes $\mathcal{G}(G, \boldsymbol{x}, u)$.*

*Proof idea.* For each tuple $\tau = \langle \mathcal{G}, G, u \rangle$, the tree structure of the corresponding fibring architecture is given by the unraveling tree of node $u$ in $G$ at the depth of $\mathcal{G}$. Indeed, a GAT computes recursively w.r.t. the depth of its unraveling tree, as do fibred neural networks w.r.t. the depth of their fibring architecture. At each node, the fibring function returns the node features of the neighboring nodes and assigns the weights of the relevant layer of $\mathcal{G}$. In the computation of the fibred neural network, the step of "replacing the entries in $S_i$ of the vector $\mathbf{x}_i$ by $\langle \mathcal{N}_i, \mathcal{F}_i, \tilde{f}_i \rangle (\mathbf{y}_i)$" is used to concatenate the outputs of the computations of the previous layer. The concatenation of outputs at the previous layer is then aggregated (or the attention layer is computed), using the relevant weights (or attention vectors) assigned by the fibring function upstream of the concatenation. The detailed proof is provided in the Appendix A.

The same result holds for GNNs and Transformer encoders simply by removing the nodes and edges in the fibring architecture $\mathcal{F}^\tau$ that encode the GAT attention mechanism and from the fact that a Transformer encoder is a GAT applied to a complete graph with positional encoding incorporated into token features $(\mathbf{x}_\xi^0)_{\xi \in S}$.

## 5.2 GNNs, GATs and Transformer encoders as fragments of fibred logics

The following result establishes a non-uniform expressiveness result for GATs, characterized by a countably infinite family of formulas in the fragment of the fibred logic of interest.

**Theorem 5.2.** *Let $\tau = \langle \mathcal{G}, G, u \rangle$, $\mathcal{N}^\tau$, $\mathcal{F}^\tau$ and $\mathscr{G}_\tau = \{\tilde{f}_{\boldsymbol{x}}^\tau\}_{\boldsymbol{x}}$ be as in Theorem 5.1. Denote $u^\tau$ the root of $\mathcal{F}^\tau$. There exists a formula $\tilde{\varphi}^\tau$ in $\mathcal{L}_{\mathcal{F}^\tau}$ such that for each $\boldsymbol{x} = (\mathbf{x}_u)_{u \in V}$, for each fibred model $\mathcal{M}$ compatible with $\langle \mathcal{N}^\tau, \mathcal{F}^\tau, \tilde{f}_{\boldsymbol{x}}^\tau \rangle$ and $\mathbf{x}_u$, the following holds:*

$$\mathcal{M}, (\pi_{u^\tau, 1})^{-1}(\mathbf{x}_u) \models \tilde{\varphi}^\tau \quad \text{iff} \quad \mathcal{G}(G, \boldsymbol{x}, u) = \text{true}$$

*Proof idea.* It suffices to consider the fibred neural networks which reproduce the computations of $\mathcal{G}$ and to invoke the correspondence between fibred neural networks and our fibred logic. Since the formulas in our fibred logic only depend on the network at the root, and the fibring architecture, it follows that the formula does not depend on the node features.

*Proof.* Let $\tau = \langle \mathcal{G}, G, u \rangle$ be a tuple with $\mathcal{G}$ a GAT with input/output dimensions $n/1$, $G$ a graph and $u$ a node in $G$. For each $\boldsymbol{x}$ node features for $G$, consider a fibred neural network $\tilde{\mathcal{N}}_{\boldsymbol{x}}^\tau = \langle \mathcal{N}^\tau, \mathcal{F}^\tau, \tilde{f}_{\boldsymbol{x}}^\tau \rangle$ obtained from Theorem 5.1.

By Theorem 4.5 applied to $\mathcal{N}^\tau$ and $\mathcal{F}^\tau$, there exists a propositional formula $\varphi^\tau$ such that for each fibred model $\mathcal{M}$ compatible with $\tilde{\mathcal{N}}_{\boldsymbol{x}}^\tau$ and $\mathbf{x}_u$, $\mathcal{M}, (\pi_{u^\tau, 1})^{-1}(\mathbf{x}_u) \models \psi(\varphi^\tau, \mathcal{F}^\tau)$ iff $\tilde{\mathcal{N}}_{\boldsymbol{x}}^\tau(\mathbf{x}_u) > 0$. Furthermore by Theorem 5.1 $\tilde{\mathcal{N}}_{\boldsymbol{x}}^\tau(\mathbf{x}_u) > 0$ iff $\mathcal{G}(G, \boldsymbol{x}, u) = \text{true}$, which gives us the result. $\square$

The same result also holds for GNNs and Transformer encoders by replacing the fibring architecture $\mathcal{F}^\tau$ obtained from Theorem 5.1. In particular, to incorporate positional encodings for Transformers, it suffices to replace $\pi_{u,1}$ by the bijection $\pi_{\xi,1}$ from worlds in $\mathbf{m}_{\xi,1}$ to $\{pos(t, s), 1 + pos(t, s)\}^n$ where $t$ is the position of token $\xi$ and $s$ is the length of the sequence.

## 6 Discussion, Conclusion and Future Work

For a given instance $\mathcal{G}$ of a neural architecture (a GNN, a GAT or a Transformer encoder), our non-uniform expressiveness results provide a countable family of formulas that characterize the network instance. We note that the result of Cuenca Grau et al. (2025) on bounded GNNs also implies a non-uniform expressiveness result in terms of first-order logic formulas: to construct a countable family of formulas that characterize a GNN $\mathcal{G}$, we can for example enumerate for $k \in \mathbb{N}$ the first-order logic formula of the bounded GNN obtained by applying $\mathcal{G}$ to all graphs with degree $k$.

Closing the gap to uniform expressiveness requires collapsing, for each instance $\mathcal{G}$, the corresponding family into a single formula in another logic. With the uniform expressiveness result of Benedikt et al. (2024), we already know that, for GNNs, the family must collapse to a Presburger formula. The

problem remains open for GATs and Transformer encoders. In particular, for Transformer encoders only a lower bound of the resulting logic is known (Barcelo et al., 2024).

We now present ideas towards the unification of uniform expresiveness results using fibring. Consider the countable family of fibred neural networks characterizing a GNN, given in the proof of Theorem 5.1. These fibred neural networks follow a common recursive structure, and by applying them to all possible input vectors $\mathbf{x} \in \{0, 1\}^n$, we can collect at each component (Kripke models of compatible fibred models) the vectors attained by the computations of the fibred neural networks, following the procedure in the proof of Proposition 4.3. Given the correspondence between the structure of the fibring architectures and the layers of the GNN (as seen in the proof of Theorem 5.1), the set of vectors collected by this procedure is closely related to the $\ell$-*spectrum* of the GNN, defined in Benedikt et al. (2024) as the set of all vectors attainable by the GNN at the $\ell$-th layer when applied to all possible inputs. In particular, the finiteness of the $\ell$-spectrum is a key argument used in the derivation of their uniform expressiveness results. An idea would thus be to identify commonalities between the fibred formulas for different inputs associated to the same GNN, in order to map them to the single Presburger formula. Many technical details must be made precise to close this gap, and we leave this research as future work. We believe, however, that the definition of fibring provided here will help systematize this investigation. Repeating the exercise to find commonalities in the fibred formulas for GATs and Transformer encoders constitutes, in this way, a new approach towards deriving uniform expressiveness results. The fibring formalism is expected to provide a unified lens, although non-uniform at this point, on the logical expressiveness of different network architectures, and might even hold the promise of offering a common methodology to uncover expressiveness results in the future.

The intuition of fibred logics applied to neural architectures, viewing neural networks as combining underlying logics, is appealing and similar in spirit to the current trend in interpretability research which tries to break down network computations into several modular reasoning steps, e.g.(Ameisen et al., 2025; He et al., 2025). We believe that scrutinizing the commonalities between fibred formulas for typical inputs, thereby reverse-engineering the fibred logical theories learned by neural networks, constitutes an exciting new take on the future possible extraction of interpretable logical rules capturing neural network's most relevant computations. Therefore, we posit that our approach to logical expressiveness via fibring may share common ground with interpretability research and hope that this paper will serve as a foundation in this direction. Furthermore, achieving uniform expressiveness results using the fibring formalism could enable new results on formal verification by leveraging existing results on the complexity of modal and fibred logics, e.g. (Wu et al., 2015).

This paper is the first to establish the exact correspondence between fibring neural networks and fibring modal logics. The formalism has allowed us to derive non-uniform logical expressiveness results for several modern neural architectures. A gap remains unresolved to bridge non-uniform and uniform expressiveness results, and we hope our work will inspire further research in this direction. We think that fibring as a formalism has the potential to enable the unification of expressiveness results for various network architectures, which were initially derived using different methodologies, with future applications in interpretability and verification.

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

# A    PROOF OF THEOREM 5.1

Let $\tau = \langle \mathcal{G}, G, u \rangle$ where $\mathcal{G} = \langle \{\mathbf{A}^\ell\}_{\ell=1}^L, \{\mathbf{B}^\ell\}_{\ell=1}^L, \{\mathbf{b}^\ell\}_{\ell=1}^L, \{\mathbf{a}^\ell\}_{\ell=1}^L \rangle$ is a GAT instance, $G = (V, E)$ is a graph with maximum degree $m$ and $u$ is a node in $G$.

Define $\mathcal{N}^\tau$ as the two-layer linear network whose first layer is the concatenation of $m + 1$ identity matrices (with no bias) and the second layer is the concatenation of the matrix $\mathbf{B}^L$, and $m$ times the matrix $\mathbf{A}^L$ (with bias $\mathbf{b}^L$).

**Definition A.1.** Let $G = (V, E)$ be a graph. A *lazy walk* in $G$ is a finite sequence of nodes $(u_0, ..., u_k)$ such that for each $i \in \{1, \ldots, k\}$ $u_{i-1} = u_i$ or $(u_{i-1}, u_i) \in E$. A lazy walk differs from a *path* in that one can stay on the same node multiple times.

**Definition A.2.** Let $G = (V, E)$ be a graph, $u \in V$, and $L \in \mathbb{N}$. The (lazy) unravelling of node $u$ in $G$ at depth $L$ is the graph that is the tree having:

- a root $u$

- a node $(u, u_1, \ldots, u_\ell)$ for each lazy walk $(u, u_1, \ldots, u_\ell)$ in $G$ with $1 \leq \ell \leq L$, and

- an edge between $(u, u_1, \ldots, u_{\ell-1})$ and $(u, u_1, \ldots, u_\ell)$ when $u_{\ell-1} = u_\ell$ or $(u_{\ell-1}, u_\ell) \in E$ (assuming that $u_0$ is $u$).

The fibring architecture $\mathcal{F}^\tau$ is constructed from the lazy unravelling of $u$ in $G$ at depth $L$ to which we add (in order to encode the attention mechanism) some nodes and edges in the following way: for each $\ell \in \{1 \ldots L\}$ (assuming that $u_0$ is $u$), each node $(u, u_1, ..., u_{\ell-1})$ has an extra child node $v_\ell$, and we require that each node $v_\ell$ has no child node. Note that by this construction, each node $(u, u_1, ..., u_\ell)$ has $|\mathrm{N}(u_\ell)| + 2$ children nodes (namely the nodes $(u, u_1, ..., u_\ell, w)$ for $w \in \{u_\ell\} \cup \mathrm{N}(u_\ell)$, and the node $v_{\ell+1}$).

Let $\boldsymbol{x}$ be node features for $G$.

The label of the edge between $(u, u_1, \ldots, u_{\ell-1})$ and $(u, u_1, \ldots, u_\ell)$ is defined by layer number 1 and the set of positions designed so the final output vector is the concatenation of the children computations. Its fibring function is as follows: on input the node feature $\mathbf{x}_{u_\ell}^0$, it returns the concatenation of vectors $\mathbf{x}_w^0$ for $w \in \{u_\ell\} \cup \mathrm{N}(u_\ell)$ and a network with the relevant weights of layer $L - \ell$ in $\mathcal{G}$ designed so that, in the computation of the fibred network, the network $\mathcal{N}_{(u, u_1, ..., u_\ell)}$ applied to the concatenation of vectors $\mathbf{h}_w^{L-\ell+1}$ for $w \in \{u_\ell\} \cup \mathrm{N}(u_\ell)$ returns the concatenation of $\mathbf{A}^{L-\ell} \cdot \sigma(\mathbf{h}_w^{L-\ell+1})$ (and $\mathbf{B}^{L-\ell} \cdot \sigma(\mathbf{h}_w^{L-\ell+1})$ when $w = u_\ell$).

The label of the edge between $(u, u_1, \ldots, u_{\ell-1})$ and $v_\ell$ is defined by layer number 2 and the set of positions is all positions of layer 2. Its fibring function takes as input a vector $\mathbf{y}$ and returns the same vector $\mathbf{y}$ (*self-fibring*) and a network using the relevant attention vector $\mathbf{a}^{L-\ell}$ (and the bias) of layer $L - \ell$ designed so that when $\mathbf{y}$ is the concatenation of vectors $\mathbf{y}_w^{L-\ell}$ for $w \in \{u_\ell\} \cup \mathrm{N}(u_\ell)$, then $\mathcal{N}_{v_\ell}(\mathbf{y})$ is the sum of vectors $\alpha_{u_\ell w}^{L-\ell} \cdot \mathbf{y}_w^{L-\ell}$ (plus the bias $\mathbf{b}^{L-\ell}$).

The neural architectures labelling nodes have the dimensions and nonlinearities required to realise the networks described above (in particular, the neural architectures labelling nodes $v_\ell$ uses the hardmax nonlinearity), and it is easy to see that the set of neural architectures labelling nodes, layer numbers, and sets of positions labelling edges are fully determined by $\tau$ (in particular, they do not depend on $\boldsymbol{x}$).

As a result, since $v_\ell$ is a leaf node, it follows that, in the computation of the fibred network, the output vector at node $(u, u_1, \ldots, u_{\ell-1})$ is given by $\mathcal{N}_{v_\ell}(\mathbf{y})$ where $\mathbf{y}$ is the concatenation of the children computations, i.e. from nodes $(u, u_1, \ldots, u_{\ell-1}, u_\ell)$.

We prove by (descending) induction on the depth that the fibred network $\tilde{\mathcal{N}} = \langle \mathcal{N}^\tau, \mathcal{F}^\tau, \tilde{f}_{\boldsymbol{x}}^\tau \rangle$ verifies $\tilde{\mathcal{N}}(\mathbf{x}_u^0) = \mathbf{h}_u^L$.

*Base ($L - 1$):* By construction, each child computation is given by the concatenation of $\mathbf{A}^1 \cdot \mathbf{x}_w^0$ (and $\mathbf{B}^1 \cdot \mathbf{x}_w^0$) for $w \in \{u_{L-1}\} \cup \mathrm{N}(u_{L-1})$, where $u_{L-1}$ is a neighbour reached at depth $L - 1$ starting from the root $u$. Applying $\mathcal{N}_{v_L}$ to the concatenation of these children computations yields the sum of vectors $\alpha_{u_{L-1}w}^1 \cdot \mathbf{A}^1 \cdot \mathbf{x}_w^0$ (plus the bias $\mathbf{b}^1$), which is $\mathbf{h}_{u_{L-1}}^1$.

*Induction step* $(\ell \to \ell - 1)$*:* By inductive hypothesis, the output vector at each node $(u, u_1, \ldots, u_\ell)$ is $\mathbf{h}_{u_\ell}^{L-\ell}$ where $u_\ell$ is a neighbour reached at depth $\ell$ starting from the root $u$. Computing the output at node $(u, u_1, \ldots, u_{\ell-1})$: by construction, each child computation is thus given by the concatenation of $\mathbf{A}^{L-\ell+1} \cdot \sigma(\mathbf{h}_w^{L-\ell})$ (and $\mathbf{B}^{L-\ell+1} \cdot \sigma(\mathbf{h}_w^{L-\ell})$) for $w \in \{u_{\ell-1}\} \cup \mathrm{N}(u_{\ell-1})$. Applying $\mathcal{N}_{v_\ell}$ to the concatenation of these children computations yields the sum of vectors $\alpha_{u_{\ell-1} w}^{L-\ell+1} \cdot \mathbf{A}^{L-\ell+1} \cdot \mathbf{x}_w^{L-\ell}$ (plus the bias $\mathbf{b}^{L-\ell+1}$), which is $\mathbf{h}_{u_{\ell-1}}^{L-\ell+1}$.

