# OpenReview forum: "From Neural Networks to Logical Theories: The Correspondence between Fibring Modal Logics and Fibring Neural Networks"
_ICLR.cc/2026/Conference — ICLR 2026 Poster_

### Official Review · Reviewer_k2Zz · 2025-10-29

**Soundness:** 2
**Presentation:** 2
**Contribution:** 3
**Rating:** 6
**Confidence:** 2

**Summary:**

Fibring of neural networks was introduced as a neurosymbolic framework to integrate learning and reasoning within neural networks. This paper bridges the gap between the fibring of neural networks and the fibring of modal logics by formalizing fibred models that are compatible with fibred neural networks. Building on this foundation, the authors derive results on the non-uniform logical expressiveness of Graph Neural Networks (GNNs), Graph Attention Networks (GATs), and Transformer encoders.

**Strengths:**

The paper establishes the connections between the fibring of neural networks and the fibring of modal logic. It provides a rigorous formalization linking fibred logic to fibred neural networks and demonstrates how this theory can be applied to widely used neural architectures, including GNNs, GATs, and Transformers, to enhance interpretability.

**Weaknesses:**

Although the paper is solid, some details are not explained clearly, making the overall presentation rather obscure. For instance, an illustrative figure for the fibred network could help clarify the concept. Please see the detailed questions below.

**Questions:**

1. In line 191, what is meant by the “accessibility relation”? Could the authors provide a formal definition or illustrative examples to clarify this concept?
2. The authors claim that non-uniform logical expressiveness results can be derived from several modern neural architectures. I wonder whether these models need to be pre-trained on specific datasets.
3. Furthermore, what specific logical expressions can be derived from GNNs, GATs, and Transformers? Could the authors provide more illustrative examples or figures to demonstrate this derivation process?

---

> ### Author Response · Authors · 2025-11-17
>
> We thank the reviewer for their positive assessment and for taking the time to write a thoughtful review.
>
> **Comment:** For instance, an illustrative figure for the fibred network could help clarify the concept.
>
> **Answer:** We thank the reviewer for their suggestion and we have added a diagram to the paper (Figure 1 page 4) that illustrates the fibring between two architectures. We hope this improves the intuition and the readability of the definition.
>
> **Comment:** "In line 191, what is meant by the “accessibility relation”? Could the authors provide a formal definition or illustrative examples to clarify this concept?"
>
> **Answer:** Thank you for pointing this out: it is jargon from modal logic literature, and we have added a footnote (l. 217 in our new submission) to clarify what it refers to. In a Kripke model, there is a set of worlds in which to interpret the formula, and an accessibility relation $R$ can be any relation on the set of worlds. It can be used to define a modality $\Box$ whose semantics would be, for example, $w \models \Box p$, meaning that proposition $p$ holds true at any world $w'$ which is related to $w$ through $R$.
>
> **Comment:** The authors claim that non-uniform logical expressiveness results can be derived from several modern neural architectures. I wonder whether these models need to be pre-trained on specific datasets.
>
> **Answer:** Our expressiveness results are general enough to apply to any instance of these architectures, i.e. any choice of weights (therefore, trained on any dataset). Note, however, that the semantics of the corresponding formulas will be different for each instance, since the semantics is captured by the evaluation of the corresponding fibred neural network. This is exactly what we touch upon in the discussion section (l.472 in our old submission and l. 498 in our new submission): studying the semantics of these formulas on *typical* inputs (rather than generic ones) could be a path towards identifying the logical theories relevant to real-world data. In other words, expressiveness tells us about what we *can* express in the most general case, but we might only care about a tiny portion of the expressive power in real-world scenarios.
>
>
> **Comment:** Furthermore, what specific logical expressions can be derived from GNNs, GATs, and Transformers? Could the authors provide more illustrative examples or figures to demonstrate this derivation process?
>
> **Answer:** To give the simplest non-trivial illustrative example, consider a two-layer GNN (layers $\ell_1$ and $\ell_2$) applied to a graph with only two nodes $u$ and $v$, and with nodes features of dimension 1 associated to a proposition $p$. Then, following Definition 4.4 (l.293 in our old submission and l. 321 in our new submission), our derivation process assigns a modality to each combination of layer/node and the corresponding formula is given by: $\Box_{\ell_2, u} ( \Box_{\ell_1, u} p \land \Box_{\ell_1, v} p ) \land \Box_{\ell_2, v} ( \Box_{\ell_1, u} p \land \Box_{\ell_1, v} p)$.

---

### Official Review · Reviewer_7tsZ · 2025-10-30

**Soundness:** 3
**Presentation:** 2
**Contribution:** 2
**Rating:** 4
**Confidence:** 3

**Summary:**

This paper formally connects fibring of modal logics and fibring of neural networks, addressing a gap in the literature. It introduces fibred models compatible with neural networks and derives non-uniform logical expressiveness results for GNNs, GATs, and Transformers. The long-term goal, according to the authors, is to enable fibring as a tool for interpreting neural networks using computational logic.

**Strengths:**

- The paper presents a novel and rigorous mathematical proof connecting fibring modal logic and fibring neural networks, filling a gap in the literature.
- It extends this framework to GNNs, GATs, and Transformers, which are widely used architectures in modern AI.
- The theoretical foundation is solid, with clear definitions and formalizations of fibred models for neural networks.

**Weaknesses:**

- The practical relevance of connecting fibring neural networks to fibring modal logic is unclear. What are the advantages of this connection for real-world applications or further research?
- Fibring neural networks are a niche approach in neuro-symbolic integration and are rarely used in practice. This raises concerns about the applicability of the paper’s findings.
- The goal of connecting fibring neural networks and GNNs is not well-articulated. The connection feels fuzzy, and a clearer presentation is needed to justify its importance.
- Theorems 5.1 and 5.2 prove the existence of a fibred neural network equivalent to a given GNN but provide no practical insights on how to identify or construct such a network. Additionally, the advantages of using the equivalent fibred neural network over the original GNN are unclear.
- The paper claims its findings apply to Transformer architectures, but the discussion focuses on GNNs and GATs. While GATs and Transformers share connections, this is insufficient to generalize the results to Transformers.
- The paper is purely theoretical, and the connection between fibring logic, fibred neural networks, and GNNs remains abstract. How can these results be applied in practice? Experiments or case studies would strengthen the paper’s impact.

**Questions:**

- What are the practical advantages of establishing a connection between fibring neural networks and fibring modal logic? Could the authors provide examples or use cases where this connection is beneficial?
- The connection between fibring neural networks and GNNs is described as fuzzy. Could the authors provide a clearer explanation or additional examples to strengthen this link?
- Theorems 5.1 and 5.2 discuss the existence of equivalent fibred neural networks. Could the authors provide guidance or methods for identifying or constructing these networks in practice?
- How could the theoretical results be applied in real-world scenarios? Could the authors include experiments or simulations to demonstrate the practical implications of their findings?
- The paper suggests that the results apply to Transformers, but the discussion focuses on GNNs and GATs. Could the authors clarify or expand on how their findings extend to Transformer architectures?
- The paper derives non-uniform logical expressiveness results for GNNs, GATs, and Transformers. Could the authors clarify what non-uniform expressiveness entails and how it differs from uniform expressiveness in this context?
- The authors mention a long-term goal of using fibring to interpret neural networks. Could they provide a roadmap or examples of how this interpretation might work in practice?

---

> ### Author Response · Authors · 2025-11-17
> **Response to Reviewer 7tsZ**
>
> We thank the reviewer for taking the time to write a thoughtful review.
>
> **Comments:** "The practical relevance of connecting fibring neural networks to fibring modal logic is unclear. What are the advantages of this connection for real-world applications or further research?" and "The paper is purely theoretical, and the connection between fibring [...] and GNNs remains abstract. How can these results be applied in practice? Experiments or case studies would strengthen the paper’s impact." and "What are the practical advantages of establishing [...] fibring networks and fibring modal logic? Could the authors provide examples or use cases where this connection is beneficial?" and "Theorems 5.1 and 5.2 discuss the existence of equivalent fibred neural networks. Could the authors provide guidance or methods for identifying or constructing these networks in practice?" and "How could the theoretical results be applied in real-world scenarios? Could the authors include experiments or simulations to demonstrate the practical implications of their findings?" and "The authors mention a long-term goal of using fibring to interpret neural networks. Could they provide a roadmap or examples of how this interpretation might work in practice?"
>
>
> **Answer:** We understand the reviewer's concern about the practical relevance of this work. We acknowledge that the applicability of our results/theory is not immediate. However, we would like to emphasize that the same can be said about most of the results in the literature on logical expressiveness of neural networks, including the works we cite (e.g. Barcelo et al. 2024, Benedikt et al. 2024). So to answer these concerns, we would like to first re-emphasize what we believe is the goal of this area of research, before answering the specific questions. Logical expressiveness tackles the question of what are the logical languages that capture the computations of GNNs, Transformer encoders etc. We believe that the underlying motivation is that logic is inherently more interpretable and verifiable that black-box machine learning systems. Once the relevant logics have been identified, this opens the way to tackling interpretability and verification with a new toolbox (computational logic), possibly leveraging existing results and algorithms on these logics to perform rule extraction, verification, etc.
>
> For example, in our introduction, we mentioned (l. 82 in our old submission and l.83 in our new submission) a verification problem, "given an output and a GNN, is there an input to the GNN that yields that output?" that is being studied through the lens of computational logic: this kind of verification problems are very relevant in practice; for example, considering a real-world scenario with a harmful output, an algorithm solving this verification problem would be able to identify inputs susceptible to generate this output. Therefore, deriving the logics encoded by modern architectures and utilising algorithms from computational logics applicable to these logics is an avenue to perform verification: this is what we touch upon in our discussion section (l. 477 in our old submission and l. 503 in our new submission).
>
> In summary, the practical relevance is not immediately apparent, and we cannot yet perform meaningful experiments, because most results on logical expressiveness are not (yet) strong enough to yield algorithms which are directly applicable, but the endeavour of studying logical expressiveness is well-motivated and practical in spirit, which we tried to reflect in both our introduction and our discussion section.
>
> With that in mind, to answer your specific questions, the relevance of our work is to make a contribution towards deriving new logical expressiveness results, the advantage of fibred neural networks is to propose a unification of two threads of logical expressiveness research, thereby opening new ways to improving current results. Finally, a (speculative) roadmap towards using fibring to interpret neural networks would be as follows: 1) close gaps between non-uniform and uniform expressiveness using fibring (by identifying commonalities between input-dependent formulas), 2) study the semantics of (uniform) formulas, again looking for commonalities between typical inputs seen at inference, 3) derive an algorithm to extract typical logical patterns in these formulas, 4) study these logical patterns against human understanding of tasks at hand to identify human-understanble systematic patterns, thereby forming human-understandable interpretation.
>
> We appreciate that this last part is very speculative, but we would like to re-emphasize an argument that motivates this potential approach and arguably a lot of the work in the field of neurosymbolic AI: since logic is inherently human-understandble, approaching interpretability directly from the lens of formal logic could perhaps facilitate the identification of human-understandable systematic patterns.

---

> ### Author Response · Authors · 2025-11-17
> **Response to Reviewer 7tsZ (2nd part)**
>
> **Comment:** Fibring neural networks are a niche approach in neuro-symbolic integration and are rarely used in practice. This raises concerns about the applicability of the paper’s findings.
>
> **Answer:** We agree with the reviewer that fibred neural networks are currently a niche in the neurosymbolic AI literature. We believe this is precisely due to the lack of a strong theory linking fibred neural networks to logic. Until now, the connection with logic was merely an intuition. In this paper, we make this intuition concrete by providing a formal connection. We hope that this will open the way to popularizing fibred neural networks and that researchers can build on this foundation.
>
> **Comments:** "The goal of connecting fibring neural networks and GNNs is not well-articulated. The connection feels fuzzy, and a clearer presentation is needed to justify its importance." and "The connection between fibring neural networks and GNNs is described as fuzzy. Could the authors provide a clearer explanation or additional examples to strengthen this link?"
>
> **Answer:** We acknowledge that the connection between GNNs and fibred neural networks may have been buried in the introduction and in the background section of the "Application to GNNs" section. We have done two modifications to the structure to make it more explicit.
>
> First, we have marked a clear distinction between the paragraph exposing the existing results on logical expressiveness of GNNs (l. 69 in our revised submission) and the paragraph presenting our contributions (l. 98 in our revised submission). The goal of connecting fibred neural networks and GNNs appears clearly: if fibred neural networks can reproduce GNNs, and fibred neural networks can be described in terms of fibred modal logics, then fibred neural networks provide a new lens into the logical expressiveness of GNNs.
>
> Second, we have also highlighted the paragraph presenting the scope of the technical results on GNNs, GATs and Transformer encoders (l. 398 in our revised submission, which was already there but as a remark l. 372 in our old submission). In particular, this paragraph explains that the results are stated for GATs but can be read for GNNs. They easily extend to GNNs by removing the attention mechanism. After each result, we explain how the same result for GNNs can be recovered (for Theorem 5.1, l. 407 in our old submission and l. 432 in our new submission; and for Theorem 5.2, l.432 in our old submission and l. 458 in our new submission). With that in mind, the connection between fibred neural networks and GNNs is formally and precisely established: indeed, Theorem 5.1 for GNNs would read:
>
> "For every tuple $\tau = \langle \mathcal{G}, G, u \rangle$, where $\mathcal{G}$ is a GNN with input/output dimensions $n/1$, $G = (V, E)$ a graph,  and $u \in V$, there exist a fibring architecture $\mathcal{F}^\tau$, a network instance $\mathcal{N}^{\tau}$ of the root architecture and a family $ \mathscr{G}_{\tau} = (\{f^\tau_x\})_x$ of collections of fibring functions matching $\mathcal{F}^\tau$, indexed by possible node features $x = (x_v)$,
>  such that, for all $x$, the fibred neural network $\langle \mathcal{N}^\tau, \mathcal{F}^\tau, f^\tau_x \rangle$ applied to $\mathbf{x}_u$  computes $\mathcal{G} (G, x, u)$."
>
>  and the proof is the same but removing the nodes and edges from $\mathcal{F}^\tau$ that encode the attention mechanism (l.615 in our old submission and l. 669 in our revised submission).
>
>  See the response below for the rationale of stating results for GATs instead of GNNs and Transformer encoders.

---

> ### Author Response · Authors · 2025-11-17
> **Response to Reviewer 7tsZ (3rd part)**
>
> **Comments:** "The paper claims its findings apply to Transformer architectures, but the discussion focuses on GNNs and GATs. While GATs and Transformers share connections, this is insufficient to generalize the results to Transformers." and "The paper suggests that the results apply to Transformers, but the discussion focuses on GNNs and GATs. Could the authors clarify or expand on how their findings extend to Transformer architectures?"
>
>  **Answer:** Similarly, we want to address this concern with our paragraph "Scope of technical results" (l.398 in our new submission) which emphasizes restrictive assumptions on the Transformer encoder architecture. Our results extend to our specific definition of Transformer encoder architecture with hard attention, which can be obtained from our GAT architecture simply by adding positional encodings. Please note that this is in line with restrictive assumptions taken in other works on logical expressiveness of Transformer encoders (e.g. Barcelo et al. 2024). We stated the precise definition of Transformer encoder architecture (l. 358 in our old submission and l. 383 in our new submission) to which our results directly apply precisely to avoid overclaiming. Similarly to GNNs, we provided, after each result, the exact steps required to extend our results to Transformer encoders (for Theorem 5.1, l.409 in our old submision and l. 434 in our new submission; and for Theorem 5.2, l. 433 in our old submission and l. 459 in our new submission).
>
>  Finally, we want to explain our rationale for stating results for GATs instead of GNNs or Transformer encoders whose theorems would read almost identically. Following our definitions, GNNs can be recovered from GATs by removing one thing, the attention mechanism, and Transformer encoders can be recovered from GATs by adding one thing, positional encodings, so our rationale was that the mental step to go from one to the other would be easier in this way (whereas it would require two mental steps to derive the result for GNNs if the results were stated for Transformer encoders for example, which feels less trivial). But if the reviewers find it confusing and would like to see something else, we are open to modifying the submission again.
>
>
>  **Comment:** The paper derives non-uniform logical expressiveness results for GNNs, GATs, and Transformers. Could the authors clarify what non-uniform expressiveness entails and how it differs from uniform expressiveness in this context?
>
>  **Answer:** We had explained the distinction between non-uniform and uniform expressiveness in the introduction (l. 70 in our old submission and l. 71 in our new submission), but we acknowledge that it was drowned in the literature review so we believe highlighting the paragraph on "Logical expressiveness of modern architectures" as we did in our new submission (l. 70) makes it more apparent to the reader. We have also modified the wording of this part to make it crisper. To summarize, uniform expressiveness provides a single formula for each instance of an architecture (independent of the input), whereas for non-uniform expressiveness, the logical formulas depend on the input. In that sense, uniform expressiveness is the ultimate goal of the research area on logical expressiveness as it provides a single formula that fully characterizes a GNN or a GAT or a Transformer encoder. As explained in the discussion section (l. 446-468 in our old submission, and l. 471-494 in our new submission) , non-uniform expressiveness results like the ones derived in our paper make steps towards this more important goal: indeed, studying the commonalities between input-dependent logical formulas is a clear path towards deriving uniform expressiveness results.
>
>
> Finally, this is loosely related to your concerns on the presentation, we have added a diagramatic illustration of fibring between two architectures to improve the presentation of our paper as requested by Reviewer k2Zz.

---

### Official Review · Reviewer_mpQY · 2025-10-31

**Soundness:** 3
**Presentation:** 2
**Contribution:** 3
**Rating:** 6
**Confidence:** 2

**Summary:**

The paper makes two key theoretical contributions. First, it establishes the connection between fibred neural networks and fibring modal logics. Specifically, the formulation relies on showing that evaluating the fibred network on an input is equivalent to checking the truth value of a formula in the fibring logic. Next, they apply this equivalence to derive logical expressiveness of well-known deep models such as GNNs, GATs and transformers. In particular, they focus on non-uniform logical expressiveness, i.e., is there a logical formula that coincides with with the output of the deep model depending on the input. The proof relies on showing that there exists a fibred network that produces the same input as the Graph Neural Network. Thus, using  the equivalence established between fibred neural networks and firing modal logic, the non-uniform expressiveness can be established.

**Strengths:**

Strengths
+ Seems to be the first work to make the prove non-uniform expressiveness results for well known Deep models including transformers
+ discovering the connection between fibring logics to prove logical expressiveness results could have implications on future theoretical results for more general logical expressiveness results which could be significant for interpretable AI.
The paper seems to make novel theoretical contributions that can prove useful in understanding why and how models like transformers can perform logical reasoning.

**Weaknesses:**

Weakness
- While this is a theoretical paper, it would have been nice to provide some application benefits that could potentially result from these theoretical advances.
- In terms of the presentation, it will be helpful to have maybe a summary of results that are known in logical expressiveness and what remains open problems, and how the proposed results addresses some of these. Section 6 describes this but it would be easier to understand the significance with a concise summary of the contributions.

**Questions:**

What are the practical implications of showing uniform vs non-uniform expressiveness of models? How would they potentially be useful in practice?

---

> ### Author Response · Authors · 2025-11-17
>
> We thank the reviewer for their positive assessment and for taking the time to write a thoughtful review.
>
> **Comment:** While this is a theoretical paper, it would have been nice to provide some application benefits that could potentially result from these theoretical advances.
>
> **Answer:** We refer the reviewer to the first part of our response to Reviewer 7tsZ, in which we discuss at length to address the concern of practical relevance.
>
> **Comment:** In terms of the presentation, it will be helpful to have maybe a summary of results that are known in logical expressiveness and what remains open problems, and how the proposed results addresses some of these. Section 6 describes this but it would be easier to understand the significance with a concise summary of the contributions.
>
> **Answer:** We had written a summary of existing results (l.69-92 in our old submission) and of our contributions (l.94-106 in our old submission) in the introduction, but we acknowledge that this was drowned in the literature review in our introduction, so we have added titles to our paragraphs (l. 70, and l. 98 in our revised submission) to make it more visible. We have also added a sentence at the end of the "Logical expressiveness of modern architectures" paragraph to emphasize the lack of a unified theory. In summary, our paper makes contributions towards solving an open problem in the literature on logical expressiveness of modern architectures: unifying the theories for GNNs and Transformer encoders (modal logics for GNNs on the one hand, and circuit complexity for Transformer encoders on the other hand). We can reasonably expect that unification would provide new avenues for extending existing results (e.g. lifting some restrictive assumptions), as the unified theory promises to be more general.
>
> Finally, this is unrelated to your questions but we have added a diagramatic illustration of fibring between two architectures to improve the presentation of our paper as requested by Reviewer k2Zz.

---

### Official Review · Reviewer_ouqW · 2025-11-02

**Soundness:** 3
**Presentation:** 3
**Contribution:** 3
**Rating:** 8
**Confidence:** 2

**Summary:**

This paper presents a formal correspondence between fibring modal logics and fibring neural networks and makes the following key contributions:
1.  It defines the concept of compatible fibred models and proves that fibred neural networks can be interpreted as logical formulas in this framework
2. It establishes non-uniform expressiveness results for Graph Neural Networks, Graph Attention Networks, and Transformer encoders.
3. It suggests using fibring as a potential unifying formalism for studying logical expressiveness of different neural architectures.

**Strengths:**

1. **Relevance**
The paper addresses a critical challenge: formally characterize the logical expressivity of various neural network architectures

2. **Significance and Novelty**
To the best of my knowledge, the proposed approach is novel in its problem formulation, the formal correspondence  between fibring modal logics and fibring neural networks, and the important theoretical results established.  Fibring could potentially be very significant and serve as a unifying formalism for studying logical expressiveness of different neural architectures.

3. **Soundness and Presentation**
Overall, the paper is well organized and well written. The key theoretical results are formally proven.

**Weaknesses:**

I could not find any major weaknesses.

One minor error in the formal definition of the recursive computation of a fibred neural network: in line 169, "N^{l_{i−1}:l_i} (**y_{i−1}**)" should be replaced with   "N^{l_{i−1}:l_i} (**h_{i−1}**)". In order words,  h_{i-1} should be used instead of y_{i-1}. y_{i-1} is the input of the instance network of u_{i-1}. It is the output of that instance network combined with x_{i-1} (i.e., h_{i-1} ) that should directly be used to compute x_i

**Questions:**

N/A

---

> ### Author Response · Authors · 2025-11-17
>
> We thank the reviewer for their positive assessment, and for their comments.
>
> **Comment:** "One minor error in the formal definition of the recursive computation of a fibred neural network..."
>
> **Answer:** Thank you for noticing this typo that we have corrected in the new revision. You are exactly right, $\mathbf{y}_i$ is used as an input for the child network, and $\mathbf{h}_i$, which is obtained from $\mathbf{x}_i$ by replacing the values of some of its entries by the output of the child network, is used for the propagation in the parent network. We have also added a diagramatic illustration of the concept as requested by Reviewer k2Zz.

---

### Author Response · Authors · 2025-12-01

Dear AC,

We would like to highlight that, in our rebuttal and the new upload of our paper, we have aimed to address the main concerns raised, particularly those regarding the presentation and organization of the paper, as well as the discussion of the practical significance of this theoretical work.


We thank again the reviewers for their time and constructive feedback.

---

### Meta-Review · Area_Chair_Fex5 · 2026-01-04

**Summary:**

Reviewers raised several concerns, including the practical relevance of the analysis, unclear aspects of the results, and the quality of the presentation. I acknowledge the importance of logic-based analyses for neural networks, which have been widely studied in the neurosymbolic (NeSy) literature. Although some concerns regarding the significance of this particular contribution remain, as discussed in the “Reviewer Concerns” section, I believe that the overall contribution is worthy of publication. I therefore recommend acceptance of the paper.

**Reviewer Concerns:**

The major concern raised by **Reviewer 7tsZ**:

> The practical relevance of connecting fibring neural networks to fibring modal logic is unclear. What are the advantages of this connection for real-world applications or further research?

remains outstanding, in my opinion, even after the rebuttal. As the authors note, studying logical expressiveness is well motivated, and I agree with this point. However, the main focus itself of this paper, the connection between fibring modal logics and fibring neural networks, is not sufficiently well motivated. As a result, the significance of the findings presented in this work is not convincing at this stage.

Other concerns, including those related to the presentation of the paper, have been addressed in the authors’ response and the revised manuscript.

**Reviewer Scores:**

**Reviewer ouqW** did not raise any major concerns. Therefore this reviewer is likely to maintain the score.

**Reviewer mpQY** raised concerns about the practical relevance of the contribution and the quality of the presentation. The authors attempted to address the issue of practical relevance. However, I believe it has not been fully resolved. As a result, this reviewer is likely to maintain the original score.

**Reviewer 7tsZ** raised several concerns, including the benefits of this connection for real-world applications, the applicability of the findings, the lack of an explicit construction of neural networks, and the limited generalization to transformer models. The authors provided detailed responses. While there might have been a chance for a score increase with further discussion between the reviewer and the authors, I believe the issue of practical relevance remains insufficiently addressed. Therefore, this reviewer is likely to maintain their negative score.

**Reviewer k2Zz** raised a minor concern regarding the presentation, which has been addressed in the revision. Accordingly, this reviewer is likely to maintain their score.

---

### Decision · Program_Chairs · 2026-01-26

Accept (Poster)